# HDL Triglycerides: A New Marker of Metabolic and Cardiovascular Risk

**DOI:** 10.3390/ijms20133151

**Published:** 2019-06-27

**Authors:** Josefa Girona, Núria Amigó, Daiana Ibarretxe, Núria Plana, Cèlia Rodríguez-Borjabad, Mercedes Heras, Raimon Ferré, Míriam Gil, Xavier Correig, Lluís Masana

**Affiliations:** 1Vascular Medicine and Metabolism Unit, Research Unit on Lipids and Atherosclerosis, Sant Joan University Hospital, Universitat Rovira i Virgili, IISPV, 43201 Reus, Spain; 2Spanish Biomedical Research Centre in Diabetes and Associated Metabolic Disorders (CIBERDEM), 28029 Madrid, Spain; 3Biosfer Teslab, 43204 Reus, Spain; 4Department of Electronic Engineering, Universitat Rovira i Virgili, IISPV, 430007 Tarragona, Spain

**Keywords:** HDL triglycerides, HDL cholesterol, HDL particle number, ^1^H-NMR, cardiovascular risk

## Abstract

While cholesterol content in high-density lipoproteins (HDLs) is a well-established inverse marker of cardiovascular risk, the importance of HDL–triglyceride (HDL-TG) concentration is not well known. We aim to examine plasma HDL-TG concentrations, assessed by ^1^H-NMR, in patients with metabolic diseases and their association with classical biomarkers. In this cross-sectional study, we included 502 patients with type 2 diabetes or metabolic syndrome attending the lipid unit of our University Hospital. The presence of arteriosclerotic plaques was assessed by ultrasonography. A complete lipoprotein profile was performed by ^1^H-NMR (Liposcale test). HDL-TG was strongly positively correlated with total triglycerides, glycerol, and fatty liver index, while a strong negative correlation was observed with HDL-cholesterol (HDL-C) and HDL-particle number (HDL-P). HDL-TG was associated with all triglyceride-rich lipoprotein parameters and had an opposite association with HDL-C and HDL-P. It was also significantly correlated with circulating cholesterol ester transfer protein (CETP). HDL-TG concentrations were higher as metabolic syndrome components increased. HDL-TG was also higher with worsening glucose metabolism. Patients with carotid plaques also showed higher HDL-TG. In contrast to HDL-C, HDL-TG is directly associated with metabolism and arteriosclerotic vascular alterations. HDL-TG should be considered a biomarker of metabolic and cardiovascular risk and could be a marker of HDL dysfunction.

## 1. Introduction

High-density lipoproteins (HDLs) are a complex metabolic system that, among other functions, removes cholesterol from macrophages within artery walls, transporting it to the liver for excretion [1]. This pathway is referred to as reverse cholesterol transport and is considered to interfere with the development of atherosclerotic lesions [2]. At the clinical level, HDL is assessed by its cholesterol (HDL-C) content. Many epidemiological studies have shown that high HDL-C plasma concentrations are associated with lower cardiovascular risk [3,4]. HDL is synthesized both in the liver and the intestines and is also formed during the lipolysis of triglyceride-rich lipoproteins. Native HDL is an Apo A1-phospholipid complex that acquires cholesterol from peripheral cells by interacting with cholesterol channels such as ABCA1 and ABCG1, according to its maturation state. Lecithin cholesterol acyltransferase (LCAT) esterifies the free cholesterol helping HDL maturation. Other proteins, such as cholesterol ester transfer protein (CETP) and phospholipid transfer protein (PLTP), determine HDL composition by interchanging triglycerides and phospholipids with other lipoprotein families. Ultimately, a spectrum of HDL particles with different sizes and composition exists [5]. HDL constitutes more than 95% of all circulating lipoprotein particles.

Despite the strong evidence of the association between low HDL-C and cardiovascular risk, HDL-C raising drugs such as niacin, fibrates, and CETP inhibitors have failed to decrease cardiovascular risk when tested in patients on statin therapy [6,7,8,9,10,11,12]. Only, some subgroups, such as patients with diabetes and high triglycerides and low HDL-C, seem to obtain a marginal benefit of fenofibrate therapy [13]. The reasons for this observation are several; different drugs have different effects on HDL composition beyond increasing HDL-C, resulting in HDL with different functional capacities [14]. Some studies have shown that HDL functionality is a better determinant of HDL cardiovascular protection than HDL-C, but tests to assess HDL functionality are not clinically available [15].

Patients with diabetes, obesity, or metabolic syndrome due to insulin resistance tend to have low HDL-C because of lower lipoprotein lipase activity and triglyceride enrichment. CETP interchanges triglycerides and cholesterol between very low-density lipoproteins (VLDL) and HDL. In the presence of high triglyceride levels, the equilibrium is displaced to cholesterol impoverishment and triglyceride enrichment. This also leads to a higher number of smaller HDL particles with higher catabolic rates, contributing to even lower HDL-C. HDL from diabetic patients is functionally defective due to particle composition alterations [16]. Our group showed that small HDLs that are triglyceride-enriched have a severely altered structure due to a “herniation” of core triglycerides to the HDL surface [17]. Recent data have shown that contrary to HDL-C, HDL triglycerides (HDL-TG) are a marker of increased cardiovascular risk [18].

Despite hypertriglyceridemia being a frequent clinical situation associated with increased cardiovascular risk, its impact on HDL function has not been evaluated and triglyceride HDL content is not measured at clinical settings. New analytical methods focused on lipoprotein particle number, size, and composition, such as ^1^H-NMR, could improve our understanding of lipoprotein metabolism alteration beyond lipid concentrations.

In this article, we aim to evaluate HDL-TG measured by NMR and their association with metabolic alterations of patients with metabolic syndrome and type 2 diabetes and subclinical arteriosclerosis data.

## 2. Results

Demographic, clinical, anthropometric, and standard biochemical characteristics of patients are shown in Table 1. The median age of the study subjects was 61 (52–67) years and 50.9% were female. Type 2 diabetes was present in 78,1% and metabolic syndrome in 80.1% of participants. Carotid atherosclerotic plaque was present in 32.8% of subjects. Cholesterol and triglyceride content, particle number, and size of HDL and other lipoprotein classes determined by ^1^H-NMR are also shown in Table 1. 

Table 2 shows the correlations between HDL-TG, HDL-C, and HDL-P with the main clinical and biochemical variables and lipidomics. A strong positive correlation between HDL-TG and total triglycerides, glycerol, and fatty liver index (FLI) was observed, while a strong negative correlation was observed between HDL-C and HDL-P. In lipidomics, while VLDL-C, IDL-C, VLDL-TG, IDL-TG, LDL-TG, and VLDL-P were positively strong correlated with HDL-TG, they were all strongly negatively correlated with HDL-C. Interestingly, HDL-TG correlations were in general opposite those of both HDL-C and HDL-P. 

Interestingly, linear regression analysis showed that HDL-TG was positively associated with total triglycerides levels (R^2^ = 0.391, *p* < 0.001) (Figure 1A) and CETP activity (R^2^ = 0.130, *p* < 0.001) (Figure 1B). The presence of type 2 diabetes and metabolic syndrome was associated with higher HDL-TG and lower HDL-C concentrations (Figure 2). A direct and positive association was found between HDL-TG and the number of metabolic syndrome components (Figure 2A), as well as with glucose metabolism status (Figure 2B) (*p* < 0.001). In contrast, and as expected, glucose status was negatively associated with HDL-C (*p* < 0.001). HDL-TG concentrations were higher in patients with carotid plaques (*p* < 0.05), while HDL-C was negatively associated (*p* = 0.05) (Figure 3). 

## 3. Discussion

We report the clinical value of measuring triglyceride content in HDL. HDL-C is a parameter inversely associated with cardiovascular risk. In metabolic alterations such as obesity, metabolic syndrome, or type 2 diabetes, HDL-C levels are lower and are interpreted as a signal of defective reverse cholesterol transport, leading to increased cardiovascular risk. On the other hand, high HDL-C is associated with cardiovascular protection although recent data suggest that this beneficial effect seems to be lost at very high HDL-C concentrations [19]. The failure of HDL-C raising drugs to prevent cardiovascular events has highlighted the complex nature of HDL metabolism that probably cannot be properly estimated by its cholesterol content. HDL is the main circulating lipoprotein in terms of particle number, representing approximately 95% of all plasma lipoproteins. On the other hand, proteomic studies have detected more than 200 proteins associated with HDL [20] and several miRNAs have been identified in HDL [21], indicating that different subgroups of HDL particles with different functions are present in the blood. A functional assessment of one of its functions, cholesterol efflux capacity from macrophages, is considered a stronger indicator of cardiovascular risk than HDL-C [22].

HDL is a lipoprotein particle with an active metabolic interaction with peripheral cells and other lipoprotein classes in circulation. HDL interchanges proteins and lipids with VLDL and LDL mediated by CETP and PLTP. In the presence of a high amount of triglyceride-rich lipoproteins, due to CETP activity, HDL is decreased, triglycerides enriched, and cholesterol impoverished. In fact, a number of triglyceride molecules do not fit in the HDL core, thus disturbing its structure [17]. Therefore, the amount of TG in each particle could be seen as a marker of altered HDL. In fact, the total triglycerides/HDL-C ratio has been considered a marker of insulin resistance [23,24] even in normal-weight adults [25], being a better indicator than HOMA-IR for metabolic syndrome [26]. The use of ^1^H-NMR to study the lipoprotein profile allows a wider view of lipid metabolism, including HDL particle number, size, and lipid content. A recent publication identified HDL-TG as a cardiovascular risk marker in a cohort with several atherosclerotic cardiovascular diseases opposite the HDL-C role [18]. In our hands, HDL-TG was positively related to glucose and insulin levels (not to HbA 1c) underlining its direct association to glucose metabolic disturbances, opposite to HDL-C and HDL-P.

High total triglycerides are strong markers of cardiovascular risk according to data from the emerging risk factor collaboration [4]; however, their association with cardiovascular events vanishes after adjustment for covariates, particularly non-HDL cholesterol. This observation suggests that triglycerides are good risk markers but do not have a causal role in the arteriosclerosis process. Cholesterol content in triglyceride-rich lipoproteins is considered a vascular damage effector in hypertriglyceridemia states, and it is referred to as a cholesterol remnant [27,28]. These lipoproteins are considered to play an atherogenic role because their size still allows them to infiltrate the subendothelial artery layer. 

Our observations suggest that high HDL-TG, due to altered structure and function, could also be an additional factor to be taken into account to explain the increased cardiovascular risk associated with high triglycerides. Dysfunctional HDL due to TG overload could alter vascular healing mechanisms. 

In summary, we report a strong association between HDL-TG and metabolic disturbances such as diabetes, metabolic syndrome, and obesity. HDL-TG is associated with hypertriglyceridemia and could be an effector of cardiovascular risk associated with triglycerides beyond the cholesterol remnant. 

We highlight the importance of this lipoprotein parameter, which is clearly associated with metabolic disturbances and vascular lesions, suggesting a putative functional role in the development of atherosclerosis. 

## 4. Materials and Methods

### 4.1. Patients 

We included 502 patients attending the Lipid Unit of our University Hospital because of metabolic alterations such as metabolic syndrome, type 2 diabetes, or combinations of them. Metabolic syndrome was diagnosed according to ATPIII criteria and type 2 diabetes according to standard clinical criteria. Patients with severe chronic diseases or neoplasms were not included. Complete anamnesis and physical examination including anthropometry data were recorded. Fatty liver index (FLI) was calculated by using 4 variables: BMI, waist circumference, triglycerides (TGs), and γ-glutamyltransferase (GGT) [29]. A blood sample was taken after a 10 h fast. Patients were not on lipid-lowering drugs or were studied after at least 6 weeks of washout. The plasma was stored at −80 °C according to standard preservation protocols in the BioBanc of our Research Institute. The project was approved by the Hospital Ethical Committee of our research institute and was in accordance with the Helsinki Declaration of World Medical association. All patients provided written consent to participate. The project identification code of the ethics committee was 11-04-28/4proj2, and the approval date was 28 April 2011.

### 4.2. Blood Analyses

Biochemical parameters, lipids, and apolipoproteins were measured using colorimetric, enzymatic, and immunoturbidimetric assays, respectively (Spinreact, SA, Girona, Spain; Wako Chemicals GmbH, Germany; and Horiba, Montpellier, France), that were adapted to the Cobas Mira Plus Autoanalyser (Roche Diagnostics, Barcelona, Spain). CETP activity was measured using a fluorometric assay (BioVision, Milpitas, CA, USA). Lecithin–cholesterol acyltransferase (LCAT) activity was assessed using a fluorometric assay (Calbiochem, San Diego, CA, USA). HMW-adiponectin levels were assessed using a commercial ELISA kit (RayBiotech, Inc., Peachtree Corners, GA, USA).

The 2D ^1^H-NMR liposcale test was used to determine lipoprotein particle number and size [30]. We deconvoluted the methyl signal of the plasma 2D ^1^H-NMR spectra by using 9 lorentzian functions to determine the lipid concentration of the large, medium and small subclasses of the main lipoprotein classes (VLDL, LDL, and HDL), and the diffusion coefficient (DC) associated with each analytical function, which is associated with the size. Finally, we combined the lipid concentration and geometric information (DC derived particle volume) in order to quantify the number of particles required to transport the measured lipid concentration of each lipoprotein subclass. Finally, weighted average VLDL, LDL, and HDL particle sizes were calculated from various subclass concentrations by summing the known diameter of each subclass multiplied by its relative percentage of subclass particle number. HDL cholesterol and triglyceride content was also determined by using NMR based on the liposcale test.

### 4.3. Carotid Examination

A total of 316 subjects from the entire cohort underwent a vascular study with the Mylab 50 X-Vision ultrasound (Esaote, Genova, Italy) to measure the carotid intima-media thickness (IMT) in the far wall of both common carotid arteries and plaque presence. Plaques were defined as an IMT higher than 1.5 mm or a protrusion into the lumen fifty percent thicker than the surrounding IMT.

### 4.4. Statistical Analyses

Normally distributed data are expressed as mean ± SD and median (25th percentile–75th percentile) for non normal distributed data. Categorical data are expressed as frequencies (percentage). The Mann–Whitney U test or the Kruskal–Wallis test were used to evaluate differences between groups, and Spearman’s test to evaluate correlations. Association of HDL-TG with plasma triglycerides and CETP was analyzed by linear regression analysis. Statistical analyses were performed using SPSS software (IBM SPSS Statistics, version 25, Madrid, Spain). All statistical tests were two-tailed and *p* < 0.05 was taken as significant.

## Figures and Tables

**Figure 1 ijms-20-03151-f001:**
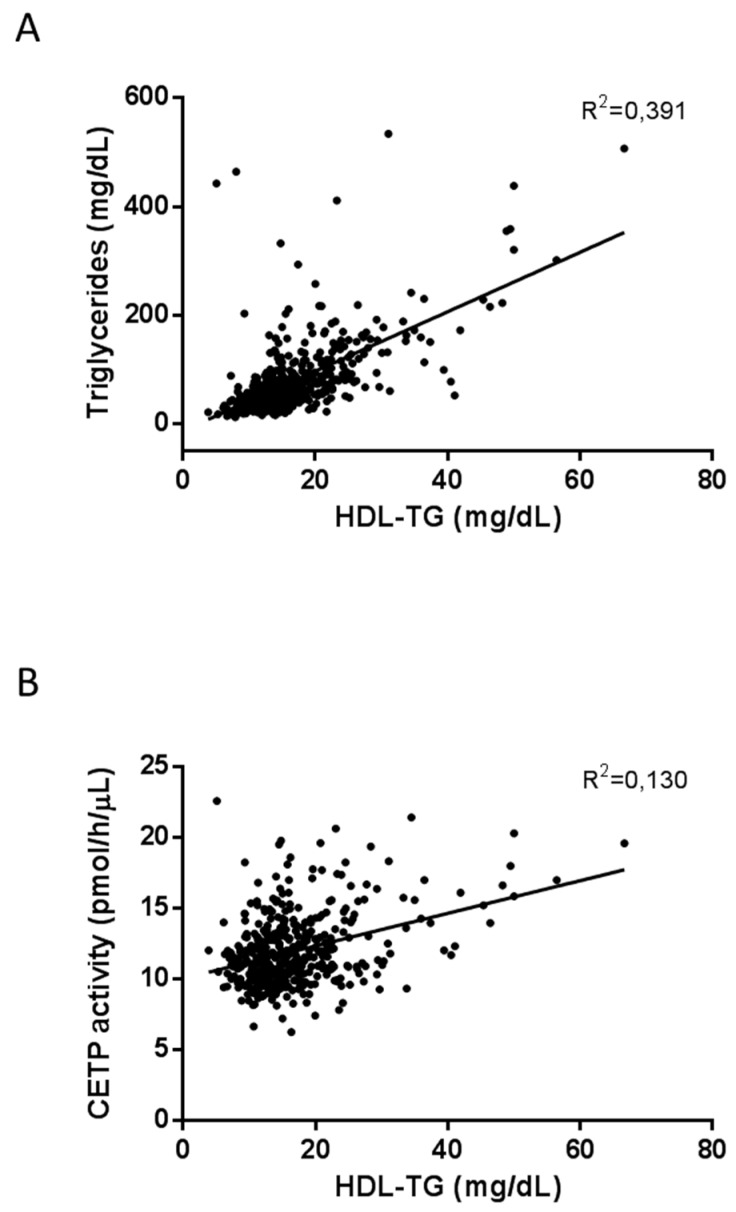
Plots of triglyceride concentrations against HDL-TG (**A**) and CETP activity (**B**). R^2^ from the regression analysis are shown.

**Figure 2 ijms-20-03151-f002:**
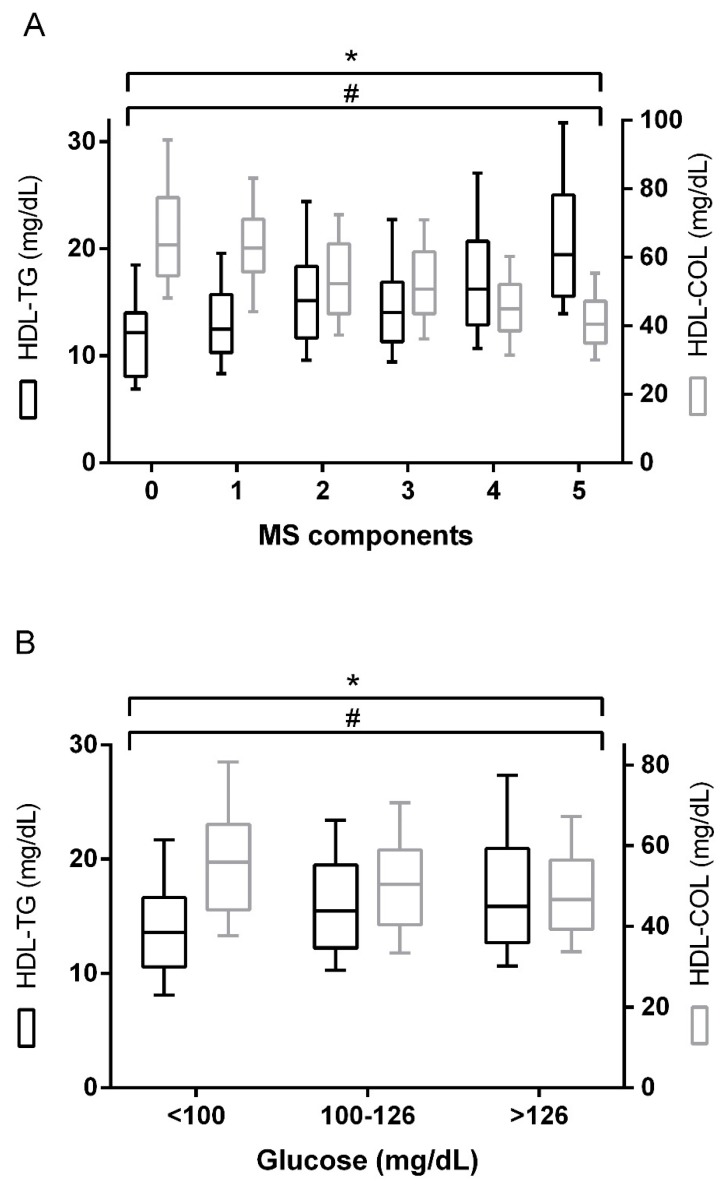
HDL-TG and HDL-C levels according to metabolic syndrome (MS) components (**A**) and diabetes status (**B**). The data are presented as box-plot. *p* values for group comparison are reported for the Kruskal–Wallis test. * *p* < 0.001 for the trend, HDL-TG; # *p* < 0.001 for the trend, HDL-C. MS components: 0 (*n* = 32); 1 (*n* = 42); 2 (*n* = 77); 3 (*n* = 126); 4 (*n* = 139); 5 (*n* = 86). Diabetes status: <100 (*n* = 112); 100–126 (*n* = 134); >126 (*n* = 256).

**Figure 3 ijms-20-03151-f003:**
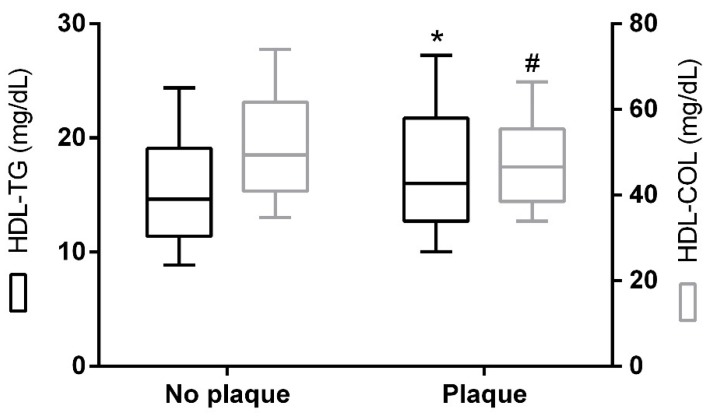
HDL-TG and HDL-C levels according to atherosclerotic plaque. The data are presented as box-plot. *p* values for group comparison are reported for the Mann–Whitney U test. * *p* < 0.05, HDL-TG; # *p* = 0.05, HDL-C. No plaque (*n* = 211); plaque (*n* = 103).

**Table 1 ijms-20-03151-t001:** Clinical covariates and lipids of the study population.

Clinical and Biochemical Variables	*n* = 502
Age (y)	61.0 (52.0–67.0)
Sex (%, women)	50.9
BMI (kg/m2)	30.0 (27.0–34.8)
SBP (mm Hg)	133.0 (124.0–146.0)
DBP (mm Hg)	80.0 (71.0–85.0)
Total cholesterol (mg/dL)	196.4 (172.8–231.2)
Total triglycerides (mg/dL)	135.9 (91.2–210.8)
ApoB100 (mg/dL)	97 (81–118)
ApoA1 (mg/dL)	138.4 ± 14.94
**Disease**	
Type 2 diabetes (%, yes)	78.1
Obesity (%, yes)	51.2
Metabolic syndrome (%, yes)	80.1
Atherogenic dyslipidemia (%, yes)	30.6
*Subclinical atherosclerosis*	
cIMT (mm) *	0.68 (0.62–0.77)
Carotid atherosclerotic plaque (%, yes) **	32.8
**Lipidomics**	
Cholesterol	
VLDL-C (mg/dL)	19.4 (10.1–34.8)
IDL-C (mg/dL)	11.2 (7.6–15.2)
LDL-C (mg/dL)	110.1 (89.6–133.7)
HDL-C (mg/dL)	49.1 (40.9–60.3)
Triglycerides	
VLDL-TG (mg/dL)	73.0 (44.5–130.8)
IDL-TG (mg/dL)	13.0 (9.7–16.2)
LDL-TG (mg/dL)	18.8 (14.5–24.0)
HDL-TG (mg/dL)	15.3 (12.2–19.5)
Particle number	
VLDL-P (nmol/L)	58.2 (33.9–102.9)
Large VLDL-P (nmol/L)	1.28 (0.82–2.10)
Medium VLDL-P (nmol/L)	4.9 (2.9–8.9)
Small VLDL-P (nmol/L)	51.7 (30.1–89.7)
LDL-P (nmol/L)	831.8 (684.8–1020.6)
Large LDL-P (nmol/L)	108.1 (89.4–132.4)
Medium LDL-P (nmol/L)	269.2 (206.4–355.3)
Small LDL-P (nmol/L)	448.9 (370.7–543.9)
HDL-P (µmol/L)	27.9 (23.8–32.2)
Large HDL-P (µmol/L)	0.28 (0.24–0.32)
Medium HDL-P (µmol/L)	8.1 (6.7–9.6)
Small HDL-P (µmol/L)	19.6 (16.3–22.7)
Particle size	
VLDL-Z (nm)	41.9 (41.8–42.1)
LDL-Z (nm)	20.9 (20.8–21.1)
HDL-Z (nm)	8.2 (8.2–8.3)

Data are shown as *n* (percentage), median (25th percentile–75th percentile), or mean ± SD. BMI indicates body mass index; SBP, systolic blood pressure; DBP, diastolic blood pressure; ApoB100, apolipoprotein B100; ApoA1, apolipoprotein A1; cIMT, carotid intima-media thickness; VLDL, very low-density lipoprotein; IDL, intermediate-density lipoprotein; LDL, low-density lipoprotein; HDL, high-density lipoprotein. * *n* = 310; ** *n* = 324.

**Table 2 ijms-20-03151-t002:** Association of HDL-TG, HDL-C, and HDL-P with clinical and lipidomic covariables in the study population.

Variables	HDL-TG		HDL-C		HDL-P	
	*ρ* (rho)	*p*	*ρ* (rho)	*p*	*ρ* (rho)	*p*
Age	0.062	0.167	0.097	0.031	0.087	0.052
SBP	0.217	<0.001	–0.201	<0.001	–0.101	0.044
DBP	0.094	0.062	–0.223	<0.001	–0.154	0.002
Waist circumference	0.145	0.002	–0.352	<0.001	–0.283	<0.001
BMI	0.157	<0.001	–0.250	<0.001	–0.183	<0.001
Cholesterol	0.233	<0.001	0.033	0.468	0.135	0.002
Triglycerides	0.652	<0.001	–0.536	<0.001	–0.197	<0.001
Apo B100	0.199	<0.001	–0.237	<0.001	–0.153	0.001
Apo A1	0.003	0.938	0.525	<0.001	0.528	<0.001
Glucose	0.183	<0.001	–0.200	<0.001	–0.102	0.022
Insulin *	0.284	<0.001	–0.367	<0.001	–0.169	0.029
HbA1c **	0.047	0.380	–0.130	0.015	–0.133	0.014
Adiponectin	–0.020	0.674	0.421	<0.001	0.362	<0.001
Glycerol	0.410	<0.001	–0.355	<0.001	–0.113	0.012
NEFA	0.212	<0.001	–0.031	0.492	0.062	0.168
CETP activity	0.264	<0.001	–0.110	0.022	0.071	0.139
LCAT activity	–0.006	0.901	0.282	<0.001	0.211	<0.001
FLI	0.363	<0.001	–0.460	<0.001	–0.268	<0.001
usCRP	0.101	0.025	–0.154	0.001	–0.112	0.013
cIMT ***	–0.023	0.682	–0.081	0.152	–0.126	0.027
VLDL-C	0.682	<0.001	–0.597	<0.001	–0.235	<0.001
IDL-C	0.534	<0.001	–0.312	<0.001	–0.136	0.002
LDL-C	–0.050	0.263	0.014	0.760	–0.072	0.105
HDL-C	–0.135	0.002		-	0.857	<0.001
VLDL-TG	0.631	<0.001	–0.607	<0.001	–0.251	<0.001
IDL-TG	0.695	<0.001	–0.337	<0.001	–0.045	0.319
LDL-TG	0.427	<0.001	–0.099	0.029	–0.008	0.851
HDL-TG		-	–0.135	0.002	0.275	<0.001
VLDL-P	0.647	<0.001	–0.608	<0.001	–0.249	<0.001
Large VLDL-P	0.622	<0.001	–0.624	<0.001	–0.266	<0.001
Medium VLDL-P	0.600	<0.001	–0.540	<0.001	–0.205	<0.001
Small VLDL-P	0.646	<0.001	–0.609	<0.001	–0.251	<0.001
LDL-P	0.047	0.291	–0.063	<0.001	–0.094	0.035
Large LDL-P	0.032	0.471	–0.041	0.354	–0.111	0.012
Medium LDL-P	–0.060	0.183	0.180	<0.001	0.042	0.350
Small LDL-P	0.110	0.014	–0.609	<0.001	–0.186	<0.001
HDL-P	0.275	<0.001	0.857	<0.001		
Large HDL-P	0.395	<0.001	0.458	<0.001	0.508	<0.001
Medium HDL-P	0.281	<0.001	0.745	<0.001	0.734	<0.001
Small HDL-P	0.234	<0.001	0.771	<0.001	0.955	<0.001
VLDL-Z	–0.271	<0.001	0.282	<0.001	0.169	<0.001
LDL-Z	–0.187	<0.001	0.427	<0.001	0.173	<0.001
HDL-Z	0.061	0.170	–0.009	0.834	–0.186	<0.001

Spearman correlations. Significance (*p* values) of rho coefficients is represented in grayscale, assuming the darkest tone as the most significant. NEFA indicates non-esterified fatty acids; CETP, cholesteryl ester transfer protein; LCAT, lecithin-cholesterol acyltransferase. *n* = 502; * *n* = 167; ** *n* = 345; *** *n* = 310.

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
