# Peer review of "HDL Triglycerides: A New Marker of Metabolic and Cardiovascular Risk"

_ijms, 2019, doi:10.3390/ijms20133151_

Reviewer 1 Report

Dear authors, thank you for the work. However, there are some points to discuss.

1.       It is known, that HbA1c represents an indicator of the average concentrations of glucose during the last 3–4 months, i.e., it characterizes also the actual glycemic compensation of diabetics in the same period. This is the reason of measuring HbA1c and considering this parameter to evaluate the glycemic compensation in medium and long terms. In 2011 the WHO Consultation concluded that HbA1c can be used as a diagnostic test for diabetes [Use of Glycated Haemoglobin (HbA1c) in the Diagnosis of Diabetes Mellitus Abbreviated Report of a WHO Consultation (2011); Davies, M.J., D’Alessio, D.A., Fradkin, J. et al. Diabetologia (2018) 61: 2461.].

However, in the present manuscript there is no data about the levels of HbA1c. Probably the data are not presented. However, according to the literature, the significant co-relations are found between HbA1c and total cholesterol, LDL and HDL, while no data of co-relations between HDL-TG and HbA1c are not known. Please, include the data of HbA1c, or discuss why they were not included.

2.       Authors should indicate whether the procedures followed were in accordance with the ethical standards of the responsible committee on human experimentation (institutional and national) and with the Helsinki Declaration of the World Medical Association with all current revisions and amendments. The institutional review board approval should be indicated in the manuscript.

Minor concerns.

Lines 36-39 “Many epidemiological studies have shown that high HDL-C plasma concentrations are associated with lower cardiovascular risk [3]. Additionally, subjects with low HDL-C have an increased risk of cardiovascular problems”. It will be good to make reference to the second sentence, or to split the both sentences.

Author Response

Dear authors, thank you for the work. However, there are some points to discuss.

1.       It is known, that HbA1c represents an indicator of the average concentrations of glucose during the last 3–4 months, i.e., it characterizes also the actual glycemic compensation of diabetics in the same period. This is the reason of measuring HbA1c and considering this parameter to evaluate the glycemic compensation in medium and long terms. In 2011 the WHO Consultation concluded that HbA1c can be used as a diagnostic test for diabetes [Use of Glycated Haemoglobin (HbA1c) in the Diagnosis of Diabetes Mellitus Abbreviated Report of a WHO Consultation (2011); Davies, M.J., D’Alessio, D.A., Fradkin, J. et al. Diabetologia (2018) 61: 2461.].

However, in the present manuscript there is no data about the levels of HbA1c. Probably the data are not presented. However, according to the literature, the significant co-relations are found between HbA1c and total cholesterol, LDL and HDL, while no data of co-relations between HDL-TG and HbA1c are not known. Please, include the data of HbA1c, or discuss why they were not included.

HbA1c was available in 89% of patients with diabetes (n=345). We have added the correlations of this variable in table 2. HbA1c was significant and negative correlated with HDL-C and HDL-P. A sentence has been added to discussion about HDL-TG associations to glucose metabolism markers (page 7, lines 29-31).

2.       Authors should indicate whether the procedures followed were in accordance with the ethical standards of the responsible committee on human experimentation (institutional and national) and with the Helsinki Declaration of the World Medical Association with all current revisions and amendments. The institutional review board approval should be indicated in the manuscript.

Sorry for that, we forgot adding this aspect. The project was approved by the Hospital Ethical Committee of our research institute and was in accordance with the Helsinki Declaration of the World Medical association. All patients provided written consent to participate. This aspect has been added to methods sections (page 8, lines25-28).

Minor concerns.

Lines 36-39 “Many epidemiological studies have shown that high HDL-C plasma concentrations are associated with lower cardiovascular risk [3]. Additionally, subjects with low HDL-C have an increased risk of cardiovascular problems”. It will be good to make reference to the second sentence, or to split the both sentences.

We have joined both sentences and also added another reference reinforcing the message (page 1, line37).

Reviewer 2 Report

This is a well performed study.  The novelty of  this paper is the emphasis on the significance of the HDL triglycerides  that is yet to be widely investigated (currently, there is quite a bit of activity in this area, including several review articles, the experimental details make this work significant and of high quality).

Perhaps the paper would benefit from quoting and  discussing the findings of the articles listed below. 

Pantoja-Torres, B., et al. (2019). "High  triglycerides to HDL-cholesterol ratio is associated with insulin  resistance in normal-weight healthy adults." Diabetes Metab Syndr 13(1): 382-388.

The  triglyceride to high-density lipoprotein cholesterol (TG/HDL-C) ratio  as a predictor of insulin resistance but not of β cell function in a  Chinese population with different  glucose tolerance status. Zhou M, Zhu L, Cui X, Feng L, Zhao X, He S, Ping F, Li W, Li Y.  Lipids Health Dis. 2016 Jun 7;15:104. doi: 10.1186/s12944-016-0270-z.

TriGlycerides  and high-density lipoprotein cholesterol ratio compared with  homeostasis model assessment insulin resistance indexes in screening for  metabolic syndrome in  the chinese obese children: a cross section study.  Liang J, Fu J, Jiang Y, Dong G, Wang X, Wu W. BMC Pediatr. 2015 Sep 28;15:138. doi: 10.1186/s12887-015-0456-y.

Author Response

Reviewer 2

This is a well performed study.  The novelty of  this paper is the emphasis on the significance of the HDL triglycerides  that is yet to be widely investigated (currently, there is quite a bit of activity in this area, including several review articles, the experimental details make this work significant and of high quality).

Perhaps the paper would benefit from quoting and  discussing the findings of the articles listed below.  

Pantoja-Torres, B., et al. (2019). "High  triglycerides to HDL-cholesterol ratio is associated with insulin  resistance in normal-weight healthy adults." Diabetes Metab Syndr 13(1): 382-388.

The  triglyceride to high-density lipoprotein cholesterol (TG/HDL-C) ratio  as a predictor of insulin resistance but not of β cell function in a  Chinese population with different  glucose tolerance status. Zhou M, Zhu L, Cui X, Feng L, Zhao X, He S, Ping F, Li W, Li Y.  Lipids Health Dis. 2016 Jun 7;15:104. doi: 10.1186/s12944-016-0270-z.

TriGlycerides  and high-density lipoprotein cholesterol ratio compared with  homeostasis model assessment insulin resistance indexes in screening for  metabolic syndrome in  the chinese obese children: a cross section study.  Liang J, Fu J, Jiang Y, Dong G, Wang X, Wu W. BMC Pediatr. 2015 Sep 28;15:138. doi: 10.1186/s12887-015-0456-y.

Thank you for your recommendation. We have added the 3 references and expanded this point in the discussion section (page7, lines25,26).

Reviewer 3 Report

In their manuscript ‘HDL Triglycerides: A New Marker of HDL Dysfunction and Cardiovascular Risk’ by Girona et al. the authors examined the potential of HDL-TG as a biomarker for cardiovascular risk during a cross-sectional study carried out in T2D/obese/metabolic syndrome patients. A major strength is the NMR method applied to analyse particle number and size. Most of the manuscript is well written and the results provide important information advancing the knowledge in the field of lipoproteins and in particular HDL as predicting risk factors/biomarkers for cardiovascular/metabolic diseases. Unfortunately however, unlike claimed by the title, HDL dysfunction has not really been investigated in this study.

Specific Comments:

1)      Can HDL dysfunction be analysed, e.g. by assessing C efflux capacity of plasma/HDL samples or vascular healing ability as mentioned in the discussion? If not this part of the title is not accurate.

2)      Introduction: HDL constitutes more than 95% of all circulating lipoproteins? This is misleading if not emphasized that it is particle number which is referred to -  or the % in terms of mass should also be indicated.

3)      Fig. 2 and 3: no n number is given. Box-Plots might be more appropriate for data presentation.

4)      Methods: diagnostic/inclusion criteria for metabolic syndrome, type 2 diabetes should be mentioned

5)      Methods: The blood was stored at -80ºC? and 4.2. Blood Analyses – should probably be plasma not blood.

Author Response

Reviewer 3

In their manuscript ‘HDL Triglycerides: A New Marker of HDL Dysfunction and Cardiovascular Risk’ by Girona et al. the authors examined the potential of HDL-TG as a biomarker for cardiovascular risk during a cross-sectional study carried out in T2D/obese/metabolic syndrome patients. A major strength is the NMR method applied to analyse particle number and size. Most of the manuscript is well written and the results provide important information advancing the knowledge in the field of lipoproteins and in particular HDL as predicting risk factors/biomarkers for cardiovascular/metabolic diseases. Unfortunately however, unlike claimed by the title, HDL dysfunction has not really been investigated in this study.

Specific Comments:

1)      Can HDL dysfunction be analysed, e.g. by assessing C efflux capacity of plasma/HDL samples or vascular healing ability as mentioned in the discussion? If not this part of the title is not accurate.

The reviewer is right, we have not assessed HDL functionality (efflux studies are in course). Therefore, we have deleted the “HDL dysfunction” concept from the title,  that is now:

HDL TRIGLYCERIDES: A NEW MARKER OF METABOLIC AND CARDIOVASCULAR RISK

 2)      Introduction: HDL constitutes more than 95% of all circulating lipoproteins? This is misleading if not emphasized that it is particle number which is referred to -  or the % in terms of mass should also be indicated.

The word “particle” has been added (page 2, line2).

3)      Fig. 2 and 3: no n number is given. Box-Plots might be more appropriate for data presentation.

We have added the number of patients in the figure legend. Furthermore, we have changed to box-plot the data presentation (page 6 and 7).

4)      Methods: diagnostic/inclusion criteria for metabolic syndrome, type 2 diabetes should be mentioned

We have added in the methods section the diagnostic criteria for metabolic syndrome and type 2 diabetes (page 8, line 19,20).

5)      Methods: The blood was stored at -80ºC? and 4.2. Blood Analyses – should probably be plasma not blood.

We have changed blood for plasma (page 8, line 24).

Round  2

Reviewer 3 Report

The authors have addressed all questions and although no additional experiments could be included they have appropriately changed text and/or data presentation.

I included two minor comments in p.2, line 2 and legend to figure ", line 3 in the attached pdf-document.
